# Growth Monitoring and Promotion Service Utilisation and Associated Factors among Children in Afar Region, Northeast Ethiopia

**DOI:** 10.3390/ijerph20105807

**Published:** 2023-05-12

**Authors:** Semhal Kiros, Ibrahim Mohammed Ibrahim, Kedir Y. Ahmed

**Affiliations:** 1Afar Regional Health Office, Logiya Primary Hospital, Semera P.O. Box 28, Ethiopia; semhalkiros24@gmail.com; 2Department of Midwifery, College of Medicine and Health Science, Samara University, Semera P.O. Box 132, Ethiopia; ibrahimmaca52@gmail.com; 3Rural Health Research Institute, Charles Sturt University, Orange, NSW 2800, Australia; 4Translational Health Research Institute, Western Sydney University, Campbelltown, NSW 2560, Australia

**Keywords:** growth monitoring, child health service utilisation, pastoral community, Ethiopia

## Abstract

The use of growth monitoring and promotion (GMP) services in the first two years of life can facilitate the early identification of common childhood health issues such as malnutrition and infections. It also creates an opportunity to promote education and nutritional counselling. This study is the first to investigate the use of GMP and its influencing factors among mothers in Ethiopia’s pastoralist regions, including the Afar National and Regional State, where childhood malnutrition is a significant cause of morbidity and mortality. Between May and June 2021, a cross-sectional study was conducted within the Semera-Logia city administration. The study used a random sampling technique to select 396 children under two, and data were gathered using an interviewer-administered questionnaire. Multivariable logistic regression was used to analyse the influence of explanatory variables, which included socio-demographic, health service, and health literacy factors, on the utilisation of GMP services. The overall utilisation of GMP services was 15.9% (95% confidence intervals [CI]: 12.0%, 19.5%). Children whose fathers had college or higher education were more likely to utilise GMP services (adjusted odd ratios [AOR] = 7.75; 95% CI: 3.01, 19.99), whereas children living in households with more children were less likely to utilise GMP services (AOR = 0.11; 95% CI: 0.04, 0.28 for households with 3–4 children and AOR = 0.23; 95% CI: 0.08, 0.67 for households with 4+ children). Children who received postnatal care had higher odds of GMP service use (AOR = 8.09; 95% CI: 3.19, 20.50). GMP services are not being fully utilised to decrease infant and child morbidity and mortality caused by malnutrition in Ethiopia. We recommend strengthening GMP services in Ethiopia and taking targeted action to address the low attainment of parental education and poor postnatal care utilisation. Public health initiatives such as the implementation of mobile health (mHealth) approaches and education of mothers by female community healthcare workers on the significance of GMP services could be effective in increasing GMP service utilisation.

## 1. Introduction

Childhood malnutrition is a top global priority, with an estimated 149 million children below the age of five years affected by stunted growth and approximately 45.4 million children experiencing wasting [1]. The consequences of malnutrition for young children are both short- and long-term, including adverse effects on cognitive and intellectual development, academic potential, and an increased risk of diseases such as diarrhea, pneumonia, and anaemia [2,3,4,5]. Additionally, the effects of malnutrition can extend to future generations, leading to reduced physical strength, decreased work productivity, unemployment, and a higher risk of poverty [6,7]. To tackle this issue, the United Nations Children’s Fund (WHO) has recommended the growth monitoring and promotion (GMP) program as a means of addressing childhood malnutrition [8].

Growth monitoring and promotion involves regularly tracking a child’s weight and height, evaluating growth against established charts, analysing the data to understand growth patterns, and taking actions such as nutritional counselling, administering supplements, or conducting medical exams to address health issues [9]. GMP has been implemented since the 1980s worldwide with the main aims of providing a diagnostic tool to promptly detect health and nutritional problems in children; promote optimal breastfeeding and appropriate complementary feeding of mothers, families, and health workers; and facilitate regular contact with primary health care facilities for infants and young children [9,10]. Theoretically, the expected benefits of GMP are achieved through improvements in nutritional status and survival by the two chief nutritional interventions (i.e., breastfeeding and complementary feeding practices), and increases in the utilisation of child health services (e.g., immunization, supplementation, and referral) using frequent health facility visits [9,10].

Despite GMP being a standard health sector practice for global children, the impact of the program on child nutritional outcomes and survival has been a subject of debate in the past decades due to the mixed evidence of success across global nations [9,10,11]. For example, evidence from Nigeria, Jamaica, Madagascar, Senegal, and Brazil documented the benefits of GMP [9,10,11,12,13], while studies from Bangladesh and India reported little or no effect of GMP [9,10,11,14]. Previously published studies have reported the main barriers to the expected success of GMP: poor provider–client communication, interventions not being tied to context-specific circumstances, skill gaps in frontline health workers (particularly in taking measurements and interpretations), shortage of time for client counselling due to patient load, and caregivers not putting health advice into practice [10,12,14,15]. In recognizing this fact, in 2020, the United States Agency for International Development (USAID) with other partner organizations revised the strategies of GMP to better integrate them with child health services in the first 12 months of birth and to maximize the promotion element within the GMP [16].

Ethiopia, the second most populous country in Africa [17,18], has made remarkable progress over the past two decades in reducing under-five mortality rates, dropping from 166 deaths per 1000 in 2000 to 67 deaths per 1000 in 2016 [19,20,21,22]. Additionally, stunting rates have decreased from 58.0% in 2000 to 37.0% in 2019. However, the fact that one in fifteen children still dies before reaching the age of five is concerning, with undernutrition remaining a major contributor to these deaths [23]. Furthermore, the Ethiopian economy continues to face several challenges, including natural and human-made disasters, such as conflicts and internal displacements, that contribute to low crop yields, food insecurity, and malnutrition [24,25,26,27].

Given the current need for maximizing the ‘promotion’ element (e.g., breastfeeding education and nutritional counselling) within the GMP and tackling the high burden of malnutrition and poor survival among children in Ethiopia [16], the Government of Ethiopia has integrated GMP with the Community-Based Nutrition Program (CBNP) [28,29]. The 2016 National Nutrition Program (NNP) has also included GMP as a potential strategy for improving the nutritional status and survival of children under two years of age [28]. Previous studies conducted at the subnational level in Ethiopia have indicated that the utilisation of GMP services was higher among certain groups of children. These groups included those who were born in healthcare facilities [30], lived in wealthier households [30], resided with fewer family members [30], had mothers with good knowledge and a positive attitude [31], had a shorter distance to healthcare facilities [31], and reported antenatal care (ANC) use [31]. 

In order to effectively address the high burden of malnutrition among pastoral mothers in the Afar region, where low agricultural productivity, frequent natural and man-made disasters, and inadequate market access infrastructure contribute to this problem [32], it is crucial to understand the barriers and drivers for implementing GMP. However, no previously published study has examined the GMP service utilisation among pastoral communities in Ethiopia to culturally integrate the GMP program by adapting to the unique needs and cultural contexts of these communities.

Furthermore, information from this study would be an input for local and global nutritional initiatives, given the 2020 NNP targets of early initiation of breastfeeding (EIBF) and exclusive breastfeeding (EBF) (80% each) [28]; the 2025 Ethiopian Health Sector Transformation Plan (HSTP-II) targets of stunting (reducing to 25%) [33]; the 2030 Global Nutrition Targets (GNTs) of EBF (increasing to 70%) and stunting (50% reduction) [34]; and the 2030 Sustainable Development Goals (SDGs) target of ending all forms of malnutrition [35]. The present study was conducted, for the first time, to investigate the utilisation and associated factors of GMP for infants and young children among pastoralist mothers in the Afar Region of Ethiopia.

## 2. Materials and Methods

### 2.1. Study Design and Study Setting

A community-based cross-sectional study was conducted to investigate the utilisation and associated factors of GMP services among children under two years of age in the Semera-Logia city administration of Afar National and Regional State (ANRS) from May to June 2021. The ANRS is geographically located in the Great East African Rift Valley System and shares a border with Eritrea (northeast), Tigray (northwest), Oromia (south), Somali (southeast), Amhara (west), and Djibouti (east) [36]. The ANRS has an estimated population size of almost 2 million with more than 12% of the population aged less than five years, and the livelihood of more than 85% of the Afar population depends on livestock production (including camel, goat, and cow) [37]. The arid or semi-arid and low rainfall climate makes the region vulnerable to adverse health problems including malnutrition, infection, and poor survival [36]. The Semera-Logia city administration is located 584 km away from the capital city of Ethiopia, Addis Ababa [38], and the city has one primary hospital, two government health centres, and seven private clinics to provide preventive, curative, and rehabilitative services.

### 2.2. Source and Study Population

The source population included all mothers/caregivers with children under two years of age who resided in the Semera-Logia city, and the study population included those who resided in the randomly selected kebeles (the lowest administrative unit in the Ethiopia context) of the administrative city.

### 2.3. Sample Size Determination and Sampling Procedure

A single population proportion formula, assuming a 95% confidence interval, 5% marginal error, and 43.9% proportion of GMP (based on a study that gave the optimum sample size) [39], was used to calculate the required sample. Considering a non-response rate of 10%, a total of 416 study participants were included in the study. To select the study participants, 3 out of 8 kebeles in the Semera-Logia city were randomly (lottery method) selected, and a list of children under two years from each kebele was pre-obtained from the health extension workers (community health workers in Ethiopia context) as a sampling frame. Finally, a random sampling technique that accounts for the total number of under two children in the selected kebeles was used to select mothers/caregivers living with them.

### 2.4. Outcome Variable

The main outcome variable for this study was GMP service utilisation, which was measured by asking mothers to recall their visits to health facilities for GMP services and the frequency of such visits. This measure was based on previously published studies conducted in low- and middle-income countries (LMICs), including Ethiopia [30,40]. GMP utilisation was defined as when a child’s height and weight measurements were recorded using a World Health Organization (WHO) standard child growth chart at least once per month, two times per 1–3 months, five times per 4–11 months, and four times per 12 months for 12–23-month-old children. For this study, the outcome variable was dichotomised as ‘Yes’ if the child received GMP at least once for less than one month, two times for 1–3 months, five times for 4–11 months, and four times per year for 12–23 months, and ‘No’ otherwise [30,40].

### 2.5. Explanatory Variables

The explanatory variables were broadly classified as sociodemographic factors, health service factors, and health literacy factors. The sociodemographic factors included child sex (grouped as ‘male’ or ‘female’), child age (grouped as ‘0–5 months’, ‘6–11 months’, ‘12–17 months’, or ‘18–23 months’), mother’s age (grouped as ‘15–24 years’, ‘25–29 years’, or ‘30–34 years’), mother’s and father’s educational status (grouped as ‘primary schooling or less’, ‘secondary schooling’, or ‘college or higher schooling’), mother’s occupation (grouped as ‘housewife’, ‘government employed’, or ‘merchant’), father’s occupation (grouped as ‘government employed’, ‘private employed’, ‘merchant’, ‘daily labourer’, or ‘pastoralist’), marital status (grouped as ‘currently married’ or ‘formerly married’), number of under-five children (grouped as ‘<3 children’, ‘3–4 children’, or ‘>4 children’), ethnicity (grouped as ‘Afar’, ‘Amhara’, or ‘Tigre/Oromo’), and wealth index (grouped as ‘poor’, ‘middle’, or ‘rich’), 

Health service factors included antenatal care (grouped as ‘yes’ or ‘no’) and postnatal care (PNC, grouped as ‘yes’ or ‘no’) visits, mode of birth (grouped as ‘vaginal delivery’ or ‘operational delivery’), and distance from a health facility (grouped as ‘<30 minutes’, ‘30–60 minutes’ or ‘>60 minutes’). Health literacy factors included awareness of the benefits of GMP (grouped as ‘yes’ or ‘no’), source of information (grouped as ‘health extension workers’, ‘other health care workers’, or ‘media’), and media exposure (grouped as ‘yes’ or ‘no’). This classification was consistent with previously published studies conducted in Ethiopia and other LMICs [30,40,41,42].

### 2.6. Data Collection and Quality Control

Data were collected using an interviewer-administered questionnaire designed to obtain information on explanatory factors (including sociodemographic health service and health literacy factors), and the outcome variable (GMP utilisation). The data collection was conducted using a Bachelor of Science Health Professional as a data collector and a Master of Health Science Professional as a supervisor. To ensure a mutual understanding of the main objectives and the data collection instrument, a one-day intensive training was provided by the primary investigator to the data collectors and supervisors. A pre-test was also performed on 5% of the study sample among mothers/caregivers in the Dubti district, and the completeness and consistency of collected questionnaires were checked every day during the data collection period.

### 2.7. Data Processing and Analysis

After ensuring the consistency and completeness of the collected data, an Epi-info version 7.2 software package was used to conduct data entry (Centers for Disease Control and Prevention, Atlanta, GA, USA), and the results were exported to SPSS version 27.0 for the final analysis (IBM Corp, Armonk, NY, USA). Principal component analysis (PCA) was used to construct a household wealth index using available information on household-level assets such as television, toilet system, and source of drinking water [43]. For this study, initial analyses involved calculating frequencies and percentages to describe the study participants, and to calculate the prevalence of GMP utilisation across the study participants. Binary logistic regression models were used to examine the influence of sociodemographic, health service, and health literacy factors on GMP utilisation. All independent variables with a *p*-value < 0.25 were entered into the final multivariable logistic regression model. Odds ratios (ORs) and 95% confidence intervals were used to report the findings of the regression modelling, and a Hosmer and Lemeshow goodness-of-fit test was used for model selection.

### 2.8. Ethical Consideration

The study ensured ethical clearance by seeking approval from the Ethical Review Committee of Samara University, College of Medical and Health Science. Written informed consent was obtained from each study participant after explaining the overall objective, the confidentiality and privacy, and the expected benefits of the study, and participants were reassured of the right to discontinuation at any stage during the interview process. All interviews were conducted with strict privacy in a quiet place, and the confidentiality of participants was kept by removing personal identifiers from the collected data.

## 3. Results

### 3.1. Sociodemographic Factors

In the current study, a total of 396 mothers/caregivers responded to the questionnaire, yielding a response rate of 95.0%. More than half (55.8%) of the children were females, and 137 (34.6%) of them were in the age group 6–11 months. Among mothers, 181 (45.7%) of them were in the age group 25–29 years, and 290 (73.2%) of mothers were Afar in their ethnicity. More than half of mothers (53.8%) attained primary schooling or less, and nearly half (49.5%) of them resided in households with 3–4 children under 5 years. The majority (81.6%) of mothers were housewives, and about 203 (51.3%) of fathers attained a high school education (Table 1).

### 3.2. Health Service Factors

Many mothers (87.4%) reported at least one ANC visit for the current baby, while only 62 (15.7%) of them visited health facilities for PNC. The majority (85.1%) of children were born using normal delivery, and about 340 (85.9%) of them reported health facility birth. One hundred and eighty-three (46.2%) mothers reported traveling more than 1 h to reach the nearest health facility (Table 2).

### 3.3. Health Literacy Factors

More than half of the study participants (55.3%) heard about the benefits of the GMP service, and, out of these, 21.7% heard about the benefits of improving health-seeking behaviour, 15.9% reported the relevance of monitoring child growth, and 7.1% reported the importance of checking the health status of the child (Table 2).

### 3.4. Utilisation of Growth Monitoring and Promotion

The proportion of GMP utilisation was 15.9% with a 95% confidence interval [CI], from 12.0% to 19.5%, and, out of this proportion, 79.2% of them reported being regular users of the GMP services (Figure 1). 

The utilisation of GMP services was higher among children aged 18–23 months compared to 12–17 months (21.7% vs. 12.2%). Mothers who attained college or higher education had a higher utilisation of GMP services compared to those with primary schooling or less (84.3% vs. 2.8%). Children whose fathers attained college or higher education had a higher utilisation of GMP services compared to those who attained primary schooling or less (56.1% vs. 8.1%). The utilisation of GMP services was higher among children who resided in rich households compared to those who resided in poor households (48.4% vs. 0.6%) [Table 1].

Children who visited health facilities for PNC had a higher utilisation of GMP services compared to those who did not visit health facilities (56.5% vs. 8.4%). Children with less than 30 min of distance from health facilities had a higher utilisation of GMP services compared to those with more than 60 min of distance (86.0% vs. 4.4%) [Table 2].

### 3.5. Factors Associated with Growth Monitoring and Promotion

Our study showed that children whose fathers attained college or higher education were more likely to utilise GMP services compared to those whose fathers attained primary schooling or less (AOR = 7.75; 95% CI: 3.01, 19.9). Children who resided in households with many children were less likely to utilise GMP services compared to those who resided in households with less than three children (AOR = 0.11; 95% CI: 0.05, 0.28 for households with 3–4 children and AOR = 0.24; 95% CI: 0.08, 0.67 for households with 4+ children). The odds for GMP service utilisation were significantly higher among mothers who reported PNC usage for the current baby compared to those who did not report PNC use (AOR = 8.09; 95% CI: 3.19, 20.5) [Table 3].

## 4. Discussion

The present study aimed to investigate the prevalence of GMP utilisation and its associated factors among children under two years of age. By conducting this study, we were able to bridge the gap in the knowledge base about the usage of GMP services and associated factors for pastoral communities. Our findings showed that the overall utilisation of the GMP services was 15.9%, highlighting the requirement for interventions to enhance their usage in the Afar Region. Children whose fathers attained college-level or higher education were more likely to utilise GMP services, while those who resided in households with many children were less likely to utilise GMP. The odds of GMP utilisation were higher among children who visited health facilities for PNC visits.

According to Ethiopia’s NNP-II (2016–2020), promoting the utilisation of GMP was one of the strategic objectives that was aimed at improving the nutritional status of children under the age of two in Ethiopia. Despite this fact, the current study showed a lower level of GMP service utilisation (15.9%) among children under two years of age in the Semera-Logia city of Afar Region, consistent with Ethiopian studies conducted in Mareka (16.9%) and Butajira (11.0%) districts. However, the current GMP utilisation was lower compared to the Lako Abaya study (39.0%) in Ethiopia, and other studies from South Africa (70%) [40], Rwanda (79%) [44], Kenya (53.3%) [41], Afghanistan 87% [45], Uganda (59%), Honduras (87%), Brazil (42%), and the Dominican Republic (85%) [10]. The study’s findings suggest that efforts should be made to strengthen GMP utilisation by maximizing the impact of the promotion element of the GMP service using evidence-based community nutrition programs.

Children living in pastoral communities face several challenges that put them at risk of malnutrition, such as inadequate food availability and quality, as well as limited access to essential services such as healthcare [32]. To tackle this issue, it is crucial to understand the unique challenges and opportunities that these communities present. One initiative that aims to promote the growth and development of children is the GMP program. However, for this program to be effective in pastoral communities, it is important to integrate it culturally to meet the needs and context of the local population. This integration process involves working closely with local communities and stakeholders, including health professionals, government officials, and community leaders, to raise awareness and build support for the program. Culturally appropriate communication strategies, such as using local languages and engaging with community influencers and leaders, should be developed to ensure that the program’s messages are clearly understood and effectively communicated.

Research conducted in LMICs shows that parents with higher levels of education are more likely to seek medical care for their infants and young children from health facilities [46]. This is because educated parents are more likely to access health-related information and have better health-seeking behaviour, leading to improved child health service utilisation, including GMP [47,48]. Additionally, educated parents may have better economic opportunities and higher household incomes, which can provide resources for adequate healthcare for their children [47]. The present study found that the educational status of fathers has a positive impact on the utilisation of GMP services for infants and young children. This is consistent with a qualitative study that emphasized the influence of husbands in decision-making regarding health service utilisation, indicating that the success of GMP in Ethiopia is dependent on the support of husbands [12]. Our finding suggests the need for improving the educational status of parents to improve the impacts of GMP services in stopping the intergenerational effects of malnutrition in Ethiopia.

The findings of this study suggest that households with a large number of under-five children in the Afar Region of Ethiopia are less likely to utilise GMP services. This is consistent with previous studies conducted in Ethiopia [30] and Indonesia [49] that have reported a similar relationship between the number of children and GMP utilisation. The relationship between the number of children and GMP utilisation can be explained through two pathways. Firstly, a higher number of children in a household may indicate a lower socioeconomic status, which can affect the ability of parents or caregivers to access healthcare services, including GMP [50]. Secondly, when the number of children increases, the resources (e.g., time) of mothers/caregivers that are available may limit their ability to seek health care. Overall, these findings highlight the need for targeted interventions to improve the uptake of GMP services in households with multiple under-five children.

Our study found that mothers who had received PNC were more likely to utilise GMP services, which is consistent with similar research conducted in the Maraka district of Ethiopia [30]. This relationship between PNC service and the increased utilisation of GMP services could be explained by the nutritional counselling and health education sessions provided as part of the service. The Sustainable Development Goal (SDG) target 3.8 aims to achieve universal health coverage and reduce healthcare inequalities between different populations, including disadvantaged groups, such as those in rural or low-income areas [35]. To achieve this target, it is important to empower households economically and to improve the health-seeking behaviour of mothers, who play a crucial role in detecting and addressing child health problems. Although the Government of Ethiopia is providing maternal and child health services free of charge in the public health facilities of Ethiopia [51], it seems that the direct and indirect costs related to time, medication, and transportation are hindering the utilisation of maternal and child health services (including GMP) among disadvantaged Ethiopian mothers.

This study has limitations. First, the cross-sectional nature of the study design makes it difficult to confer the direction of causality; nevertheless, the findings are consistent with other studies conducted in Ethiopia [30,39]. Second, a social desirability bias in assessing the awareness of mothers regarding the growth chart might be a possible limitation of the study. Despite the limitations, this study was the first to be conducted on the GMP service utilisation in the pastoral community, which supports the policymakers and practitioners working in these areas.

## 5. Conclusions

The study indicated that the utilisation of GMP services was insufficient in promoting the well-being, growth, and survival of children. Children living in households with many siblings were less likely to utilise GMP, whereas those who received PNC services at healthcare facilities and whose fathers had a higher education were more likely to utilise GMP services. To address the current GMP service utilisation gap in pastoral communities, there is a need for system- and policy-level changes. Enhancing GMP services in Ethiopia and targeting efforts to modifiable factors, such as improving parents’ educational attainment and postnatal care use, should be prioritised. Public health initiatives such as implementing mobile health (mHealth) approaches and educating mothers through female community healthcare workers on the importance of GMP services and proper nutrition could be effective in increasing GMP service utilisation. Future cultural integration of the GMP services to pastoral communities may require qualitative research on culturally appropriate health communication strategies.

## Figures and Tables

**Figure 1 ijerph-20-05807-f001:**
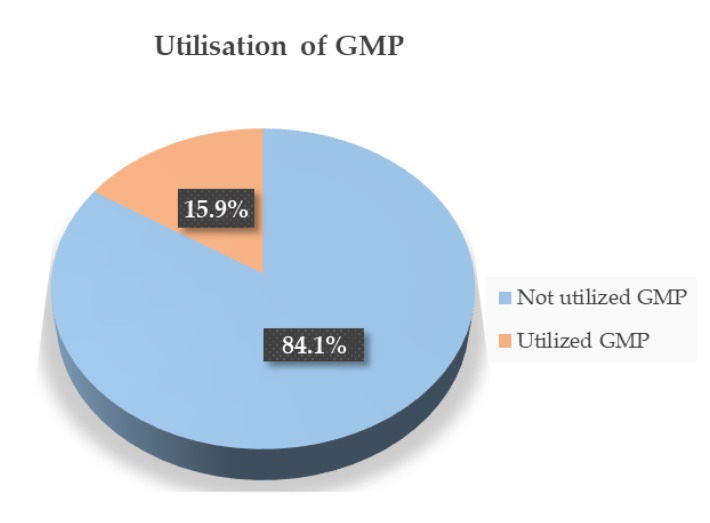
Utilisation of growth monitoring promotion service in Semera-Logia city of Afar Region, Ethiopia, 2021 (n = 396).

**Table 1 ijerph-20-05807-t001:** Sociodemographic characteristics of mothers in the Semera-Logia City, Afar Region, Ethiopia, 2021 (n = 396).

Variables	n (%)	GMP Utilisation	
		Yes, n (%)	No, n (%)
Child sex			
Male	175 (44.2)	26 (14.9)	149 (85.1)
Female	221 (55.8)	37 (16.7)	184 (83.3)
Age of the child in months			
0–5 months	67 (16.9)	12 (17.9)	55 (82.1)
6–11 months	137 (34.6)	21 (15.3)	116 (84.7)
12–17 months	123 (31.1)	15 (12.2)	108 (87.8)
18–23 months	69 (17.4)	15 (21.7)	54 (78.3)
Mother’s age in years			
15–24 years	168 (42.4)	24 (14.3)	144 (85.7)
25–29 years	181 (45.7)	34 (18.8)	147 (81.2)
30–34 years	47 (11.9)	5 (10.6)	42 (89.4)
Mother’s educational status			
Primary schooling or less	213 (53.8)	6 (2.8)	207 (97.2)
Secondary schooling	132 (33.3)	14 (10.6)	118 (89.4)
College or higher	51 (12.9)	43 (84.3)	8 (15.7)
Mother’s occupation			
Housewife	323 (81.6)	30 (9.3)	293 (90.7)
Government employed	37 (9.3)	29 (78.4)	32 (88.9)
Merchant	36 (9.1)	4 (11.1)	8 (21.6)
Marital status			
Currently married	369 (93.2)	58 (15.7)	311 (84.3)
Formerly married	27 (6.8)	5 (18.5)	22 (81.5)
Father’s educational status			
Primary schooling or less	111 (28.0)	9 (8.1)	102 (91.9)
Secondary education	203 (51.3)	8 (3.9)	195 (96.1)
College or higher	82 (20.7)	46 (56.1)	36 (43.9)
Father’s occupation			
Government employed	179 (45.2)	-	14 (100)
Privately employed	97 (24.5)	4 (6.9)	54 (93.1)
Merchant	58 (14.6)	47 (26.3)	132 (73.7)
Daily labourer	48 (12.1)	12 (12.4)	85 (87.6)
Pastoralist	14 (3.5)	-	48 (100)
Number of under-5 children			
<3 children	1111 (28.0)	47 (42.3)	64 (57.7)
3–4 children	196 (49.5)	9 (4.6)	187 (95.4)
>4 children	89 (22.5)	7 (7.8)	82 (92.1)
Ethnicity			
Afar	290 (73.2)	32 (11.0)	258 (89.0)
Amhara	60 (15.2)	20 (33.3)	40 (66.7)
Tigre/Oromo	46 (11.7)	11 (23.9)	35 (76.1)
Wealth index			
Poor	173 (43.7)	1 (0.58)	172 (99.4)
Middle	98 (24.7)	2 (2.04)	96 (98.0)
Rich	124 (31.3)	60 (48.4)	64 (51.6)

**Table 2 ijerph-20-05807-t002:** Health service and health literacy factors among mothers in the Semera-Logia City, Afar Region, Ethiopia, 2021 (n = 396).

Variables	n (%)	GMP Utilisation	
		Yes, n (%)	No, n (%)
Health service factors			
Antenatal care visits			
Yes	346 (87.4)	55 (15.9)	291 (84.1)
No	50 (12.6)	8 (16.0)	42 (84.0)
Postnatal care visits			
Yes	62 (15.7)	35 (56.5)	27 (43.5)
No	334 (84.3)	28 (8.4)	306 (91.6)
Mode of delivery			
Vaginal delivery	337 (85.1)	41 (12.2)	296 (87.8)
Operational delivery	59 (14.9)	22 (37.3)	37 (62.7)
Distance to the nearest health facility			
<30 min	50 (12.6)	43 (86.0)	7 (14.0)
30–60 min	163 (41.2)	12 (7.4)	151 (92.6)
>60 min	183 (46.2)	8 (4.4)	175 (95.6)
Health literacy factors			
Heard about the benefits of GMP			
Yes	177 (44.7)	58 (32.8)	119 (67.2)
No	219 (55.3)	5 (2.3)	214 (97.7)
Awareness of the benefits of GMP			
Seek medical care	86 (21.7)	9 (10.5)	77 (89.5)
Monitor child growth	63 (15.9)	45 (71.4)	18 (28.6)
Know health status	28 (7.1)	4 (14.3)	24 (85.7)
Source of information on GMP			
Health extension workers	30 (7.6)	28 (93.3)	2 (6.7)
Other health workers	25 (6.3)	25 (100)	-
Media	8 (2.2)	8 (100)	-
Media exposure			
Yes	358 (90.4)	63 (17.6)	295 (82.40)
No	38 (9.6)	-	38 (100)
Watching television			
Yes	323 (81.6)	63 (19.5)	260 (80.5)
No	73 (18.2)	-	73 (100)

**Table 3 ijerph-20-05807-t003:** Factors associated with GMP utilisation among children under two years of age in the Semera-Logia City of Afar Region, Ethiopia, 2021 (n = 396).

Variables	GMP Utilisation		COR (95% CI)	AOR (95% CI)	*p*-Value
	Yes	No			
Husband’s educational status					
Primary schooling or less	9	102	1.00	1.00	
Secondary schooling	8	195	0.46 (0.17, 1.24)	0.59 (0.2–1.7)	0.330
College or higher education	46	36	14.5 (6.44, 32.5)	7.75 (3.01–19.9)	<0.001
Number of under-5 children					
<3 children	47	64	1.00	1.00	
3–4 children	9	187	0.07 (0.03, 0.25)	0.11 (0.04, 0.28)	<0.001
>4 children	7	82	0.12 (0.05, 0.27)	0.23 (0.08, 0.67)	0.006
Postnatal care visit					
Yes	35	27	1.00	1.00	<0.001
No	28	306	14.2 (7.51, 26.7)	8.09 (3.19, 20.50)	
Mode of delivery					
Vaginal delivery	41	296	1.00	1.00	
Operational delivery	22	37	4.29 (2.3–7.9)	0.99 (0.32–3.02)	0.985

COR: crude odds ratio; AOR: adjusted odds ratio; operational delivery: includes C/S delivery and instrumental delivery.

## Data Availability

Restrictions apply to the availability of these data.

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
