# Peer review of "Growth Monitoring and Promotion Service Utilisation and Associated Factors among Children in Afar Region, Northeast Ethiopia"

_ijerph, 2023, doi:10.3390/ijerph20105807_

Round 1

Reviewer 1 Report

The results of Table 1 indicate that mother's educational status is also associated with GMP use, with mother's who have received a college education or higher associated with increased GMP. This is not highlighted in the discussion or conclusion. In lines 237-240 it is stated that a father's support is likely needed to improve GMP use and perhaps this needs to be amended to focus more on improving education, considering increased education of both mother's and father's is associated with increased GMP use. 

Table 1 also shows increased use of GMP by those categorized as "rich", which is not highlighted in the results or conclusion. As appropriate, perhaps the discussion and/or conclusion should be modified to suggest greater attention to decreasing socioeconomic disparities, including access to education.

Please double check for subject-verb agreement and appropriate use of capitalization. 

Reviewer 2 Report

I enjoyed reading your work "Growth monitoring and promotion service utilisation and associated factors among children in Afar Region, Northeast Ethiopia". The overall presentation is good and I appreciate to revise the paper based on the following comments

Abstract

Please interpret the conclusion in broader way e.g. what need to be done to address “ low attainment of parents education and poor postnatal care utilisation

 Introduction

Please define Growth monitoring and promotion in the introduction to be clear for none-expert reader

Line 72

The last paragraph can you please remove “water shortages” to keep the flow of the sentence

“water shortages, low agricultural production, frequent nat-72 ural and manmade disasters, and poor infrastructures to access markets”

Method

-Complete the study area in one paragraph

-The definition of Outcome variable GMP needs more clarification ie where did you check those measurements used to define GMP (self-report or record review). The approach you defined yes /no for GMP needs  more clarity

Standard child growth chart also needs to be define as there will be different option and which one is used in your context.

- Explanatory variables: if there is no word limit please provide more clarity and grouping the measurement of independent variables

Discussion

In addition to narrating the results please include a statement about the knowledge addition to the existing literature by this study.

Add the interpretation of the result in the context of pastoralist community (study population)

Conclusions

Needs rewritten by considering more context (eg system and policy level changes to address the current gap) that needs to be improved in the pastoral community e.g. mobile education and health service access.

 Do you have ethical approval  to share participant data to anyone (did participants know as you will share their data with a third party) “Data Availability Statement: The datasets used and/or analysed during the current study are avail-297 able from the corresponding author on reasonable request.”

Reviewer 3 Report

If you can provide a little more background information on cultural integration of the GMP program in Ethiopia, that would be helpful. 

You had a very impressive response rate of 95%. Did those who declined to participate provide a reason for declining? Please cite the reason if they did. Clarify if all communicated in same language/dialect since you have a global audience who may not be familiar with the intricacies of culture in Ethiopia. If 53% of the mothers have primary or less schooling, were they writing down information in the questionnaires or did the interviewer speak with them and write down their responses in the forms? I think this should be clarified in methods because it qualitatively changes the data. 

Obviously, you are bringing attention to the multifactorial nature of improving GMP utilisation. I think you also need to address transportation issues in the discussion. More than half of the respondents live 30 minutes or more from the nearest health facility. I think there are additional modifiable factors to focus on in terms of considerations of exposure to media to increase awareness of benefits of GMP utlisation and so on. 

I don't think you clarified all of the abbreviations used in the manuscript. Please make sure that they are broken down into the words that they represent the first time that they are introduced. 

Reviewer 4 Report

Manuscript is based on the study reviewing factors related to the use of growth monitoring and promotion (GMP) services in the first two years of life in the Afar region of Ethiopia. Overall, results from this study are well presented well and derived from a well-designed and conducted study.

Data were collected from mothers/fathers of 396 children under the age of two. Overall, they found that 15.9% took advantage of GMP services. Their data indicate that parents with fewer children and with higher educational attainment are more likely to participate in this program.

While their conclusions are firmly based on the results some additional thought should be given to the topics of wealth and resource availability. Data indicate that wealthier families were more likely to utilize this service. Additional analyses should be conducted within this variable as well as distance from and transportation resources available to the program. They mentioned that 46% of their population traveled  more than one hour to the healthcare facility. Is information available on distance to and, more importantly, transportation resources available?

Another consideration may be given to the definition of GMP compliance. It appears that their definition is quite rigorous involving multiple child visits over a 12 month period. I am assuming that the 15.9% participation is based on compliance across all visits. Have the authors calculated participation based on partial compliance? How many attended a one-month visit? Three-month? Etc. Is it necessary that the child be examined at each point in time?

Minor points:

p.3, line 127:  “casre” should be care

p. 8, line 234: “like” should be likely
